# Accurate Early Detection and *EGFR* Mutation Status Prediction of Lung Cancer Using Plasma cfDNA Coverage Patterns: A Proof-of-Concept Study

**DOI:** 10.3390/biom14060716

**Published:** 2024-06-17

**Authors:** Zhixin Bie, Yi Ping, Xiaoguang Li, Xun Lan, Lihui Wang

**Affiliations:** 1Department of Minimally Invasive Tumor Therapies Center, Beijing Hospital, National Center of Gerontology, Institute of Geriatric Medicine, Chinese Academy of Medical Sciences, No. 1 Dongdan Dahua Street, Beijing 100730, China; biezhixin3910@bjhmoh.cn (Z.B.); lixiaoguang4678@bjhmoh.cn (X.L.); 2Department of Basic Medical Sciences, School of Medicine, Tsinghua University, Beijing 100084, China; pingyi900@gmail.com; 3Tsinghua-Peking Joint Center for Life Sciences, Tsinghua University, Beijing 100084, China; 4Centre for Life Sciences, Tsinghua University, Beijing 100084, China; 5MOE Key Laboratory of Bioinformatics, Tsinghua University, Beijing 100084, China

**Keywords:** cfDNA, coverage patterns at the transcription start sites, early cancer screening, *EGFR* mutation status prediction, machine learning

## Abstract

Lung cancer is a major global health concern with a low survival rate, often due to late-stage diagnosis. Liquid biopsy offers a non-invasive approach to cancer detection and monitoring, utilizing various features of circulating cell-free DNA (cfDNA). In this study, we established two models based on cfDNA coverage patterns at the transcription start sites (TSSs) from 6X whole-genome sequencing: an Early Cancer Screening Model and an *EGFR* mutation status prediction model. The Early Cancer Screening Model showed encouraging prediction ability, especially for early-stage lung cancer. The *EGFR* mutation status prediction model exhibited high accuracy in distinguishing between *EGFR*-positive and wild-type cases. Additionally, cfDNA coverage patterns at TSSs also reflect gene expression patterns at the pathway level in lung cancer patients. These findings demonstrate the potential applications of cfDNA coverage patterns at TSSs in early cancer screening and in cancer subtyping.

## 1. Introduction

Lung cancer remains the leading cause of cancer-related deaths worldwide, significantly impacting global public health. The 5-year survival rate for lung cancer is only 10% to 20%, mainly due to its rapid progression and often late-stage diagnosis, leading to treatment complexity and decreased survival rates [1]. Most patients are diagnosed with advanced-stage disease (stage III or IV), with a less than 5% 5-year survival rate. More than half of lung cancer patients die within one year of diagnosis [2]. Patients diagnosed in early or mid stages have a better prognosis than those in later stages, with higher chances of successful treatment, highlighting the importance of screening for early cancer detection and improved prognosis.

Liquid biopsy, as a non-invasive technique, is not only applicable for cancer screening but also for real-time monitoring of cancer. It provides a new approach for non-invasive cancer detection. In recent years, liquid biopsy has utilized various features of cfDNA. For example, Liang et al. used deep methylation sequencing to extract cfDNA methylation data as features for machine learning classifiers, demonstrating that methylation sequencing can detect tumor-origin signals at extremely low dilution factors and establishing a model for lung cancer screening [3]. Additionally, Esfahani et al. used another feature, fragmentation entropy as an epigenomic cfDNA feature, to predict RNA expression levels of individual genes and developed the EPIC-seq tool for diagnostic, prognostic, and therapeutic use [4]. Some studies have not only used cfDNA features from liquid biopsy for cancer screening but also provided possibilities for predicting various aspects of cancer subtypes. For example, Raman et al. used copy number analysis (CNV) of cfDNA from shallow whole-genome sequencing (0.1–0.5X) to achieve histological classification of adenocarcinoma, squamous cell carcinoma, and small cell carcinoma. Through validation with paired tissue and blood, this method achieved classification without the need for tissue [5].

However, most studies on using liquid biopsy-derived cfDNA features for cancer detection and cancer status focus on genetic alterations, with limited attention to the cfDNA distribution patterns at the cis-regulatory elements. Studies have shown a close correlation between the non-random representation of cfDNA sequences and the intracellular gene regulatory landscape [6,7]. These correlations increase the possibility of inferring intracellular epigenomic and transcriptomic information from cfDNA sequences [8]. Recent studies used transcription start sites (TSSs) coverage, nucleosome footprints, and other features to infer gene expression levels, but there is currently no effective method to characterize the molecular features of different subtypes of lung cancer while screening for lung cancer.

In this study, we successfully established two models based on cfDNA TSS coverage patterns from 6X whole-genome sequencing: an Early Cancer Screening Model and an *EGFR* mutation status prediction model. We enrolled a total of 196 participants for model construction and validation, and we collected another independent cohort consisting of 142 individuals from a different hospital for further testing of our models’ performance. These models achieved consistently high performances in early-stage lung cancer screening and differentiating lung cancer patients with or without *EGFR* mutation. We also characterized the overall gene expression differences in lung cancer using TSS coverage, demonstrating that this approach is a potential and efficient method for early screening and *EGFR* mutation status predicting lung cancer. Our study serves as a proof of principle demonstrating the possibility of using TSS cfDNA distribution patterns to further reveal various clinically relevant categorizations of lung cancer, facilitating the diagnosis and companion diagnostics.

## 2. Methods

### 2.1. Patient Enrollment and Sample Collection

Plasma samples from patients diagnosed with lung cancer and healthy donors for building models were obtained from the Beijing Hospital. Ethical approval (2022BJYYEC-404-01) for sample collection and use was obtained from the Ethics Committee of the Beijing Hospital. External independent validation cohorts were obtained from the Ethics Committee of Peking University Cancer Hospital (2020KT101). Approval for sample collection and use was obtained from the Ethics Committee of the Beijing Hospital and Peking University Cancer Hospital, and informed consent was obtained from all participants. Blood samples of patients were collected from all participants aged between 18 and 92 years at their initial visit, when the diagnosis was communicated, before surgical resection or treatment. Pathological results obtained from tissue/cell biopsy were used to confirm whether the individuals had lung cancer, to decide their inclusion in our cohort, and to determine the Histological subtypes and staging of lung cancer among these patients. Table 1 and Appendix A provide the clinical data for all participants. 

### 2.2. cfDNA Sample Collection and Sequencing

The collection and sequencing of cfDNA refer to the process of obtaining and analyzing cell-free DNA (cfDNA) through a series of steps. Firstly, blood samples are collected using EDTA blood collection tubes (BD, Franklin Lakes, NJ, USA, #367525), ensuring that the tube walls are intact, the caps are sealed tightly, and the tubes are within their expiration date. Within 2 h of collection, whole blood is centrifuged at 1600× *g* for 10 min at 4 °C, and the supernatant is collected and subjected to a second centrifugation. The plasma sample is centrifuged at 16,000× *g* for 10 min at 4 °C to remove all remaining cell debris. The separated blood precipitate is stored together with the plasma and transported at −80 °C until the time of DNA extraction. Next, cfDNA extraction is performed using the TianGen Magnetic Bead Blood Genomic DNA Extraction Kit (TianGen, Beijing, China, #DP709). The plasma sample (1~2 mL) is taken into a centrifuge tube, and Proteinase K solution and lysis solution are added. The mixture is incubated at 65 °C for 15 min. Isopropanol is then added, followed by the addition of magnetic bead suspension. Impurities are removed using a two-step elution with the wash buffer provided in the kit. The tube is left at room temperature to dry for 10 min. Then, 60 µL TB elution buffer is added.

NGS cfDNA libraries were prepared for WGS using at least 5 ng of cfDNA as input material. cfDNA libraries were prepared using the NEBNext Ultra II DNA Library Prep Kit for Illumina (New England Biolabs (NEB), Ipswich, MA, USA, #E7645L), following the instructions provided in the kit manual. Briefly, adapters with a single T overhang were ligated to the end-repaired dA-tailed fragments using Ultra II Ligation Master Mix and Ligation Enhancer. After excising U bases with the USER enzyme (NEB, #M5505L), the reactions were cleaned using VAHTS DNA Clean Beads (Vazyme, Nanjing, China, #N411-03), and DNA was eluted in 17 µL of H_2_O. For PCR, each reaction was mixed with 5 μL of 10 μM N5 primer, 5 µL of 10 μM N7 primer (NEB, #E7500S), and 25 µL of NEB Next Ultra II Q5 Master Mix. Amplification was performed using the following thermocycling program: 98 °C for 30 s; 12 cycles of 98 °C for 10 s, 65 °C for 75 s; and a final incubation at 65 °C for 5 min. PCR products were cleaned, and size ranges were assessed using a (Bioptic, Taiwan, China, #Qseq100). The library preparation process is efficient, taking approximately 4 h to complete. After the construction of each sample library is completed, PE150 sequencing is performed using (BGI, Shenzhen, China, #DNBSEQ-T7), with a sequencing data volume of 20 G and a sequencing depth of approximately 6–7x, which is considered low-depth whole-genome sequencing.

### 2.3. Bioinformatics Analysis and Modeling 

Firstly, quality control and data processing of cfDNA data were conducted. FastQC was employed for meticulous quality control analysis of the FASTA files to assess their quality comprehensively. Following this, the sequencing reads underwent alignment to the human reference genome (version hg38) utilizing the bwa (Burrows-Wheeler Alignment Tool, v.0.7.17-r1188). Picard tools were utilized to sort the BAM files, remove duplicates, and enhance the reliability of downstream analyses (“Picard Toolkit.” 2019. Broad Institute, GitHub Repository, San Francisco, CA, USA. https://broadinstitute.github.io/picard/ (accessed on 14 October 2021); Broad Institute). Samtools v.1.6 was applied to add indices to BAM files, filter reads, and calculate the read depth for each sample for further analyses [9]. Specific regions within the hg38 reference genome characterized by low mappability (MAPQ < 2) were meticulously filtered out. Additionally, reads bearing specific bitwise flags were excluded using the parameters -q 2 and -F 3840.

cfDNA TSS relative coverage was calculated based on the transcription start site (TSS) locations (93,701) from ENSEMBL biomaRt grch38 for humans. Samtools was used to calculate the depth of TSS regions spanning 500 bp upstream and downstream of the TSS and the background cfDNA coverage, which was defined as the average cfDNA coverage in the 1kb upstream and downstream regions. cfDNA TSS relative coverage was defined to infer chromatin accessibility [9].
$$TSS\;relative\;coverage=\frac{\sum_{i-500}^{i+500}coverage}{1000}/\;\frac{(\sum_{i-1500}^{i-500}coverage+\sum_{i+500}^{i+1500}coverage)}{1000*2}\;\;$$*i* represents the genome locations of TSSs.

The TSS relative coverage within specific regions was further refined through normalization using the z-score derived from the training cohort. This normalization approach and data were consistently applied across the validation and independent validation cohorts. 

In this study, bioinformatics tools were employed to analyze differences between groups. Principal Component Analysis (PCA) was conducted using the factoextra package v.1.0.7 [10]. Pathway and biological process enrichment analyses were performed using the R package clusterProfiler v.4.10.1 to enrich lists of genes of interest [11]. The parameters for the enrichKEGG function were set as ‘OrgDb = org.Hs.eg.db, keyType = “SYMBOL”, pvalueCutoff = 0.05’ [12]. Heatmaps were generated using the pheatmap library. ROC curves were generated using pROC v.1.18.5 [13].

Furthermore, for the Early Cancer Detection Model and *EGFR* mutation status prediction model, the random forest was employed to differentiate different groups to reach the research goal of early cancer detection and *EGFR* mutation status detection. For all models developed in this study, the cfDNA cohort was randomly divided into training data, comprising 70% of the original dataset, and validation data, comprising 30% of the original dataset. The R package ‘caret v.6.0-94’ was implemented to perform the machine learning random forest algorithm [14]. The best parameters for early cancer detection were ‘n.trees = 1500, mtry = 1′, and ‘n.trees = 2500, mtry = 5′ for *EGFR* mutation status detection. To estimate prediction error, we conducted three iterations of ten-fold cross-validation on the training data. 

Feature selection was carried out by eliminating features with near-zero variance and high correlation based on the *p*-values (<0.01) of groups. For the early cancer detection model, we first calculated the mean of normalized TSS coverage values for features showing differences between groups in the training cohort, considering healthy individuals and each cancer stage. We selected two types of features: one type with TSS mean values that increased according to the classification of healthy, stage I, stage II, stage III, and stage IV, and another type with values that decreased according to the order of this group. After sorting based on the absolute difference in value between stage I and healthy, the top 200 sites were selected for each type of feature as input features for the model. Finally, 100 TSS sites that were higher in cancer samples than in healthy individuals and 100 TSS sites that were higher in healthy individuals than in cancer samples were selected as the feature input. 

For the *EGFR* mutation status prediction model, we identified 26 pathways containing EGFR out of 619 pathways in the Canonical Pathways gene sets from the KEGG MEDICUS pathway database [15]. The genes associated with these pathways overlapped with our dataset, resulting in 2673 features. These features were used in the training cohort to compute the *p*-values for inter-group differences, and only 65 genes with *p*-values < 0.01 were retained for use in the *EGFR* mutation status prediction model.

Accuracy was calculated for the detection of lung cancer and different stages of cancer in the training data and validation cohort. Finally, we assessed the performance of the machine learning model on both the training cohort, validation cohort, and independent validation cohort.

## 3. Results

### 3.1. Enrolled Participants

Overall, this experiment established two models, an Early Cancer Screening Model and an *EGFR* mutation status prediction model, based on cfDNA TSS data obtained from 6X whole-genome sequencing. We collected blood samples from two different cohorts of individuals in this study, one with 196 participants for modeling and validation and another with 142 samples from an external independent test cohort for further validating the model’s generalization ability.

As shown in Figure 1, the process of establishing our Early Cancer Screening Model included 196 participants, including 96 lung cancer patients and 100 healthy control samples. These 196 participants in the study cohort were randomly assigned to the training cohort (total of 138 cases, including 68 patients and 70 healthy individuals) and the Validation Dataset (total of 58 cases, including 28 lung cancer cases and 30 healthy individuals). The staging data and other clinical characteristics of the cancer patients included in the study cohort are summarized in Table 1. Our lung cancer cohort includes patients at various stages, with 32 cases of stage I, 3 cases of stage II, 22 cases of stage III, and 39 cases of stage IV. Additionally, our data includes 72 cases of LUAD, 15 cases of LUSC, 3 cases of LCNEC, and 2 cases of SCLC patients. It also includes 45 cases of patients without lymph node metastasis, 51 cases of patients with metastasis, 57 cases of patients with distant metastasis, and 39 cases without distant metastasis. Information on patients and healthy cohorts for the 196 samples is detailed in Appendix A. The healthy cohort, ranging in age from 22 to 70 years, consists of 46 females and 54 males. Meanwhile, the cancer patients range in age from 36 to 91 years, comprising 37 females, 49 males, and 13 unknowns (Appendix A).
Figure 1The cfDNA transcription start site (TSS) obtained through non-invasive testing can be subjected to Whole Genome Sequencing (WGS) to achieve Early Cancer Screening and the *EGFR* mutation status prediction for lung cancer. This study involved 196 participants, including 96 lung cancer patients and 100 healthy controls, for model building. Models were based on random forest (RF) model with 10-fold 3 times cross-validation (CV) to avoid overfitting. In addition, 142 participants were included in an external independent test cohort for further model validation. To build the Early Cancer Screening Model, the 196 samples were randomly assigned to the training cohort (n = 138) and the Validation Dataset (n = 58). The external independent test cohort was used to test the generalization ability of the model. Additionally, the feasibility of using cfDNA TSS coverage for predicting *EGFR* mutations was explored using a subset of patients with clinical information on *EGFR*. To build the *EGFR* mutation status prediction model, the 65 samples were randomly assigned to the training cohort (n = 47) and the Validation Dataset (n = 18).

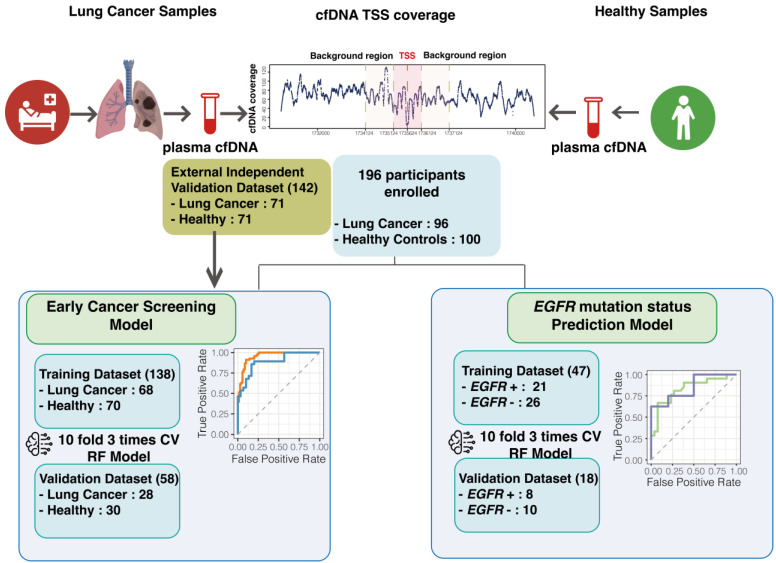

biomolecules-14-00716-t001_Table 1Table 1Clinical information of lung cancer patients. (Definition of abbreviation: LUAD (lung adenocarcinoma), LUSC (lung squamous cell carcinoma), LCNEC (lung large cell neuroendocrine carcinoma), SCLC (small cell lung cancer)).StageHistologyLymph NodesMetastasisMetastasisI32LUAD72045057II3LUSC1512121III22LCNEC321828IV39SCLC233134

NA4

42





53





71


We extracted cfDNA from the plasma of all participants and performed 6X whole-genome sequencing. We calculated cfDNA TSS coverage as a feature (method) and constructed a random forest model based on 10-fold 3-times cross-validation to build the Early Lung Cancer Screening Model.

To ensure the credibility and generalization ability of the model in real-life applications, we also included an external independent test cohort of 142 participants, including 71 lung cancer cases and 71 healthy individuals, when constructing the lung cancer early screening model.

Furthermore, based on the enrolled cohort, we also explored the feasibility of using cfDNA TSS coverage to predict *EGFR* mutations. Among the 65 patients with clinical information on *EGFR*, 47 were lung cancer patients, with 21 being *EGFR*-positive and 26 being *EGFR* wild-type. We then included 18 of these patients in the training and validation sets, with 8 being positive and 10 being negative for *EGFR* mutations. The models were all validated using ROC curves to ensure their robustness.

### 3.2. Transcription Start Site Coverage Was Distinct between Cancer and Non-Cancer Samples

The coverage values of transcription start sites (TSSs) derived from cfDNA are valuable for distinguishing between cancer and non-cancer status. This is due to the unique nature of TSSs as open chromatin regions, which are not bound by nucleosomes and, therefore, become exposed during degradation. This exposure leads to a decrease in coverage compared to surrounding regions [16]. Esfahani, M. S. et al. utilized the fragmentation entropy of promoters in this region (NDR) as an epigenomic feature to differentiate between cancer and healthy states. They also validated the correlation between NDR and RNA expression levels [4]. Moreover, TSSs may be indicative of tumor status and can be used to monitor ctDNA dynamics and disease progression [17,18]. 

Specific normalized TSS coverage values distinguishing between lung cancer and healthy individuals hold biological significance. To identify TSSs differentiating cancerous from non-cancerous samples, we selected 19,656 transcripts (*p* < 0.01) from the training cohort showing significant distinctions between cancer and non-cancer groups. A visible clustering between lung cancer and healthy individuals can be demonstrated in PCA (Figure 2A). Subsequently, we associated these 19,656 transcript features with genes to analyze their relationship with gene expression. A volcano plot of differentially expressed genes between lung cancer and healthy individuals, derived from TCGA RNA-seq data, highlighted 4245 genes with differential expressions. Among these, 1241 genes overlapped with those identified using TSS values, shown in black (Figure 2B). Notably, well-known genes linked to lung cancer development, such as CCNE1, commonly present in various tumor types when amplified at chromosome 19q12, were identified. Gallo et al. designed a new drug, RP-6306, based on this gene, which effectively kills cancer cells by inhibiting the enzyme activity of amplified cells [19]. Additionally, alterations in CDKN2A, recognized as an important tumor suppressor gene, are associated with ICB resistance in several solid tumors [20]. We utilized TCGA RNAseq data from lung cancer samples and identified only 209 differentially expressed genes between stages I/II and stages III/IV, with limited correlation to cancer progression (Appendix A). Consequently, we did not further explore TSSs to identify specific sites that distinguish between early and advanced cancer stages.

TSSs also reflect their biological significance at the pathway level. To further analyze cancer-related pathways associated with differentially expressed TSSs between cancer and non-cancer samples, we performed gene enrichment analysis on these transcripts. We selected the top 25 pathways based on significant *p*-values and gene ratio sorting. Pathways specifically related to lung cancer included “Non-small cell lung cancer” and “ErbB signaling pathway”. Other pathways related to cancer development included “TNF signaling pathway”, “One carbon pool by folate”, “Cellular senescence”, and “Bacterial invasion of epithelial cells.” These pathways directly related to lung cancer further illustrate that the selected TSS features can differentiate between lung cancer and healthy contributors and exhibit certain distinctiveness (Figure 2C).

The features selected for dimensionality reduction in the model also showed significant discriminative power. To better construct a lung cancer screening model, we further filtered the over 10,000 selected transcripts, mainly based on the differences between TSS stage groups (method). We observed differences in TSSs between healthy individuals and lung cancer patients among these filtered 200 features (Figure 2D). 

### 3.3. A Robust Early Cancer Screening Model Based on Transcription Start Sites Coverage

Utilizing the selected normalized TSS relative coverage scores, we established a robust Early Cancer Screening Model with high sensitivity, particularly for stage I samples, by using 200 biologically significant features as input. The results, as shown in the ROC curve, demonstrated consistently high performance in the training set, with an AUC of 0.966 (0.941–0.991, 95% CI). In the validation set, the AUC was 0.925 (0.860–0.991, 95% CI) (Figure 3A). 

Our model also demonstrated good prediction ability for early-stage lung cancer. To evaluate the sensitivity and specificity of detecting early-stage cancer patients using our model, we classified stage I/II as early stages and stage III/IV as advanced stages. The final model performance showed an AUC of 0.943 (0.901–0.985, 95% CI) for early-stage differentiation and 0.980 (0.960–0.999, 95% CI) for advanced-stage differentiation in the training set (Figure 3B). In the validation set, the AUC was 0.965 (0.913–1, 95% CI) for early-stage differentiation and 0.910 (0.821–0.993, 95% CI) for advanced-stage differentiation (Figure 3C). The AUC values for early stages were above 0.94 in both datasets, demonstrating the model’s good performance in early cancer screening and stable predictive ability.

To validate the model’s generalization ability, we introduced an external independent test cohort consisting of 142 samples to verify the model’s performance in distinguishing between cancer and non-cancer samples. Similarly, we obtained good model prediction results with an AUC of 0.891 (0.837–0.944, 95% CI) (Figure 3D).

We observed an age imbalance between the lung cancer and healthy groups (Appendix A). To ensure that age did not influence the performance of the Early Cancer Screening Model, we first analyzed the relationship between age and model predictor scores among the 196 samples within both cancer and healthy groups. We identified a weak correlation between age and predictor scores, with a slight decreasing trend in scores as age increased (Appendix A). Additionally, given that both the model-building cohort and external independent test cohort initially included predominantly younger healthy individuals, we also collected 16 plasma samples from older healthy individuals. To establish the aged validation cohort, we included the external 16 individuals aged 40 to 72 (Appendix A) and 7 healthy individuals aged 40 to 70 from the external independent test cohort (Appendix A). We also included 18 lung cancer patients within a similar age range (41–75) from the external independent test cohort (Appendix A). Integrating their data into the Early Cancer Screening Model, we still achieved robust predictive results (Appendix A), indicating minimal impact of age on our model’s performance.

Our dataset predominantly included cancer patients in advanced stages; therefore, we have also been monitoring patient survival data as recorded in Appendix A. We compared the model predictor scores between deceased and alive cases and analyzed the differences. Although no significant differences emerged between the groups (*p* = 0.16), the average predictor scores were higher in deceased cases than in those alive (Appendix A). 

### 3.4. Epidermal Growth Factor Receptor Mutation Status Prediction Model 

For patients diagnosed with lung cancer, particularly non-small cell lung cancer (NSCLC), the mutation status of the Epidermal Growth Factor Receptor (*EGFR*) at multiple time points is pivotal for tailoring targeted therapies and investigating resistance mechanisms [21]. Tyrosine kinase inhibitors (TKIs) like gefitinib, which target *EGFR* mutations, and crizotinib, aimed at the *ALK* gene mutations, have become integral to the precision medicine approach in treating lung cancer. Given that such patients are frequently in advanced stages of cancer, relying solely on tissue samples for mutation detection is fraught with limitations [22,23]. Consequently, various studies have underscored the potential of high-depth mutation NGS (Next Generation Sequencing) to serve as a non-invasive alternative, approximating the diagnostic precision of invasive biopsies in determining *EGFR* status [4].

In this context, TSS (transcription start site) features within circulating free DNA (cfDNA) have been identified as viable markers for distinguishing between *EGFR*-positive and *EGFR*-negative lung cancer cases. By deriving KEGG path scores for the *EGFR*-positive and *EGFR*-negative patients [15], we demonstrated that, from the perspective of EGFR-related pathways, there is a discernible segregation between *EGFR*-positive and *EGFR*-negative samples, with TSS values being particularly enriched in pathways related to EGFR (Figure 4A). Notably, 15 of these pathways exhibited significant differences (*p* < 0.01), including those crucial for the activation of *EGFR* mutations and the modulation of downstream genes. Furthermore, path scores for most *EGFR*-negative patients were observed to exceed those of their *EGFR*-positive counterparts, hinting at an epigenetic phenomenon where TSS regions, representing areas of gene activation, exhibit lower coverage. 

To validate the significance of the differential path scores derived from TSS coverage, we conducted a comparative analysis between the comprehensive pathway dataset and subsets not implicated in *EGFR* mutations. Out of the total 619 KEGG pathways analyzed, 146 (23.6%) demonstrated variances in path scores, with a striking 57.7% (15/26) of EGFR-related pathways showing significant intergroup differences (*p* < 0.01), as illustrated in Figure 4B. In contrast, among the non-EGFR mutation pathways, only 29 out of 143 (20.3%) displayed significant differences (*p* < 0.01). This disparity underscores the pronounced impact of EGFR-related pathways compared to the broader dataset (Figure 4C).

Building on these insights, we sought to establish a predictive model with a strong discriminative capability for *EGFR* mutation status. Leveraging a novel feature selection approach, we curated features for model development, focusing on those within the 26 EGFR-related pathways. Through an analysis of TSS site differences between groups, we pinpointed 65 genes (*p* < 0.01). These genes, as visualized in a heatmap, effectively distinguish between *EGFR*-positive and wild-type cases (Figure 4D). Incorporating these genes as model features yielded robust results, with the training set achieving an AUC of 0.987 and an AUC of 0.763 in the validation phase, thereby demonstrating the model’s efficacy in predicting *EGFR* mutation status with high accuracy (Figure 4E). 

The goal of conducting these studies was to demonstrate that our approach could complement traditional mutation detection methods for drug guidance, especially for patients lacking hotspot mutations but exhibiting abnormal gene expression regulation in EGFR-related pathways. Our method, based on chromatin accessibility, allows for the inference of changes in the gene expression regulation patterns of pathways by analyzing plasma cfDNA. This can guide drug selection for patients without mutations.

## 4. Discussion

Our study demonstrates the potential of utilizing transcription start site (TSS) coverage derived from circulating cell-free DNA (cfDNA) for lung cancer early screening and precise *EGFR* mutation status prediction. The Early Cancer Screening Model, based on TSS data from 6X whole-genome sequencing, showed good prediction ability for early-stage lung cancer. This suggests that TSS coverage could serve as a valuable biomarker for identifying lung cancer at an early stage. Furthermore, the *EGFR* mutation status prediction model showed high accuracy in distinguishing between *EGFR*-positive and wild-type cases. This is important for tailoring targeted therapies and investigating resistance mechanisms in lung cancer patients.

Our study also highlights the potential of TSS coverage in revealing the mutation status of lung cancer. By analyzing TSS data, we were able to identify genes and pathways that are differentially expressed within cancerous samples. This information could be used to further understand the molecular mechanisms underlying different mutation statuses of lung cancer and to develop more targeted treatment strategies.

In recent years, the technology of liquid biopsy based on plasma cfDNA has rapidly developed and improved within the field of precision oncology detection, presenting clear advantages over traditional tissue biopsy [24]. These advantages include non-invasive and convenient sampling for patients, the ability to continuously and in real-time monitor tumor progression, and the ability to effectively overcome issues related to tumor heterogeneity and multiclonality [25]. The early screening model shows excellent performance in detecting early-stage cancers, surpassing existing reported studies [4,5,26]. Given that the majority of cfDNA in plasma originates from hematopoietic cells, with tumor-derived ctDNA typically representing a very small fraction (0.01–1.0%), and that the abundance of ctDNA is influenced by tumor stage, location, or vascularization, the frequency of genetic mutations in cfDNA is low. This is particularly true for early-stage tumors, affecting the detection rate [27]. The use of TSS coverage values as a basis for classification in our study effectively avoids the impact of ctDNA abundance, as our experiment screened features across the whole genome, allowing for the identification of numerous distinctive features beyond just tumor hotspot mutations. Additionally, many TSS sites used for distinguishing cancer from non-cancer originate from tissue-specific signals in cancer tissues, providing a theoretical basis for TSSs in cancer screening [4]. The model’s generalization capability was demonstrated using an independent validation cohort. It is worth noting that while our study demonstrates the potential of TSS coverage in lung cancer diagnosis and treatment, further validation and refinement of the models are needed. Future studies should focus on validating the models in larger and more diverse patient populations, as well as exploring the utility of TSS coverage in monitoring disease progression and treatment response. In real-world clinical practice, based on traditional molecular typing methods, the prevalence of *KRAS* mutations in NSCLC ranges from 26% to 41% [28], and the prevalence of activating *EGFR* mutations in European NSCLC patients is approximately 12–15% [29]. Theoretically, our method has the potential to classify patients, thereby benefiting a broader patient population. Additionally, this method has certain advantages over other detection techniques in a clinical setting. It requires only 2ml of blood plasma and has a shorter operation time compared to probe capture methods. Moreover, the cost of library construction is lower, making it more feasible for clinical applications.

In evaluating the limitations and potential for future improvement of this study, several factors come into play. First, the small data volumes for our models support them primarily as proof-of-concept studies. For our ‘Early Cancer Screening Model’, we incorporated an external independent validation cohort of 142 subjects to further verify the model’s effectiveness. However, this external cohort lacks staging information of cancer patients, which is critical for corroborating the model’s ability to differentiate early-stage lung cancer. The incorporation of a larger cohort with comprehensive tumor-specific information would enhance the outcomes and further validate the early detection capabilities of our model. Second, both the training (N = 47) and validation (N = 18) datasets for the ‘*EGFR* mutation status prediction model’ are relatively small. Despite employing 10-fold, 3-times cross-validation to minimize overfitting, a larger dataset is essential for refining the model. Future iterations might also include a variety of *EGFR* mutation types, such as T790m, L858R, 19del, and G719, to refine the model’s details. We hope to utilize balanced cohorts with comprehensive classification information in further work to develop models that can categorize histology subtypes. Third, our data collection started recently, making it challenging to track and collect treatment responses as survival rates have not significantly changed. This limitation hinders further associative analysis between survival rates and our predictor scores. Future work will need to extend follow-ups and include tumor-specific treatment information to track survival. Additionally, integrating paired data from tissue samples with plasma data would make the experiment more comprehensive and credible.

In conclusion, our study highlights the potential of TSS coverage derived from cfDNA as a valuable biomarker for lung cancer early screening and *EGFR* mutation status prediction. This approach could lead to improved outcomes for lung cancer patients by enabling earlier detection and more personalized treatment strategies. 

## Figures and Tables

**Figure 2 biomolecules-14-00716-f002:**
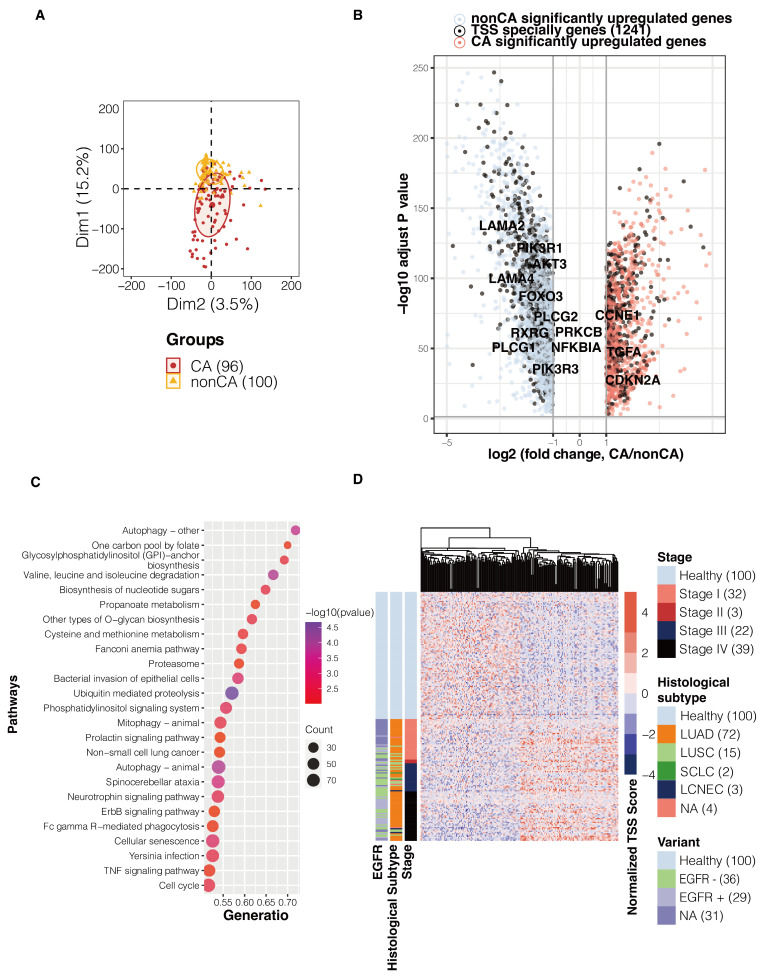
TSS coverage was distinct between cancer and non-cancer samples. (**A**) Principal Component Analysis (PCA) demonstrated clear group separation between lung cancer and healthy individuals based on TSS coverage values. (**B**) A volcano plot of differentially expressed genes between lung cancer and healthy individuals, derived from TCGA RNA-seq data, highlighted 1241 genes identified using TSS values, indicating their potential role in lung cancer development. (**C**) Gene enrichment analysis identified the top 25 pathways associated with differentially expressed TSSs between cancer and non-cancer samples, including pathways specifically related to lung cancer (e.g., “Non-small cell lung cancer” and “ErbB signaling pathway”) and other pathways related to cancer development. (**D**) Differences in TSS between healthy individuals and lung cancer patients were observed among the filtered 200 features selected for dimensionality reduction in the model, suggesting their potential utility in constructing a lung cancer screening model.

**Figure 3 biomolecules-14-00716-f003:**
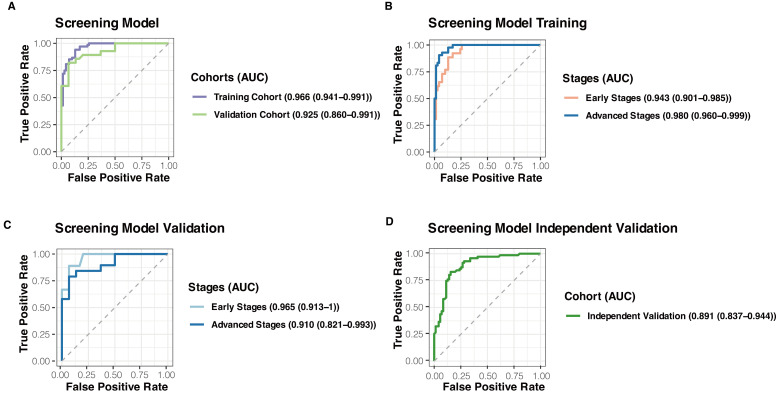
Performances of the robust Early Cancer Screening Model. (**A**) Receiver Operating Characteristic (ROC) curve demonstrating high performance of the Early Cancer Screening Model in the training and validation sets. (**B**) Area Under Curve (AUC) values for early-stage (stage I/II) and advanced-stage (stage III/IV) differentiation in the training set. (**C**) AUC values for early-stage and advanced-stage differentiation in the validation set. (**D**) Model performance in the external independent test cohort showing good generalization ability with an AUC of 0.891.

**Figure 4 biomolecules-14-00716-f004:**
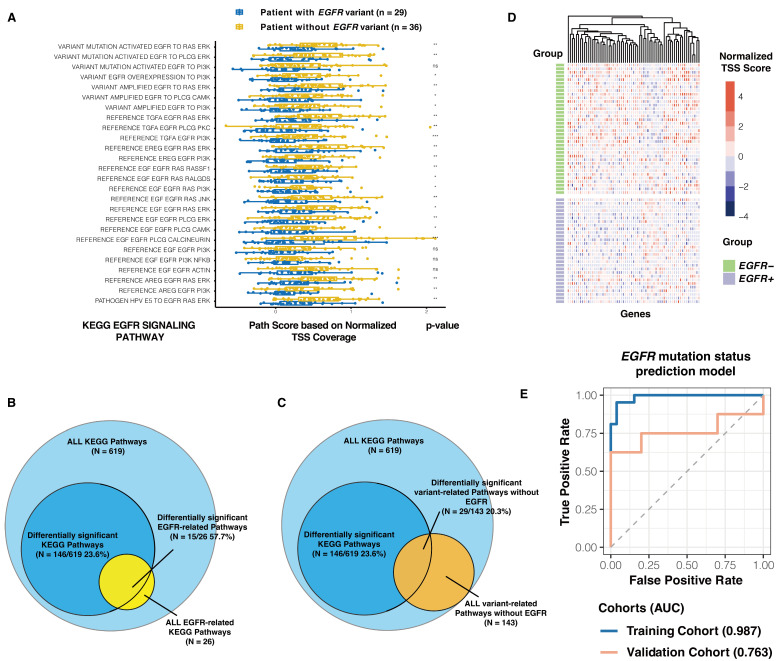
TSS coverage can differentiate patients with and without *EGFR* mutations. (**A**) 26 pathways related to EGFR from the KEGG database were selected. For each of the 65 patients with *EGFR* mutation detection, the average normalized TSS coverage of genes involved in these 26 pathways was calculated. Boxplots of path scores based on the normalized TSS coverage mean in these 26 pathways were plotted for patients with and without *EGFR* mutations, with differences between groups indicated by *p*-values (*** represents *p* < 0.001, ** represents *p* < 0.01, * represents *p* < 0.05, ns represents not significant). (**B**) Out of the total 619 KEGG pathways analyzed, 146 (23.6%) demonstrated variances in path scores, with a striking 57.7% (15/26) of EGFR-related pathways showing significant intergroup differences (*p* < 0.01). (**C**) Among the non-EGFR mutation pathways, only 29 out of 143 (20.3%) displayed significant differences (*p* < 0.01). (**D**) The TSS coverage of all transcripts corresponding to genes in the 26 pathways was calculated, and the TSS with differential expression between groups was selected for heatmap analysis, demonstrating significant differences between patients with and without *EGFR* mutations. (**E**) An *EGFR* mutation status prediction model was built and showed good performance in both the training and validation cohorts.

## Data Availability

Codes and data for this project can be accessed at https://github.com/yiping801026/cfDNA_lungcancer (accessed on 4 April 2024).

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
