# Peer review of "Accurate Early Detection and EGFR Mutation Status Prediction of Lung Cancer Using Plasma cfDNA Coverage Patterns: A Proof-of-Concept Study"

_biomolecules, 2024, doi:10.3390/biom14060716_

Round 1

Reviewer 1 Report

Comments and Suggestions for Authors

I believe this method can serve as a valuable approach for (early) lung cancer detection, however, this method fails to accurately distinguish between lung cancer subtypes and lacks tumor-specific information for treatment response. This limitation should at least be acknowledged in the discussion, as well as the feasibility to be used in a clinical setting for lung cancer detection (e.g. price tag).

Comments that need to be addressed:

Line 70-76 ‘The Early Cancer Screening Model illustrated …’: It is rather unusual to mention results in the introduction.

Line 78 ‘…and precise subtyping of lung cancer.’ (also line 280, 287, 315, 319 and 322): This cannot be stated. Figure 2D does not support this either.

Line 92 ‘These 196 participant in the study cohort were randomly assigned to the training and validation cohort.’: I am missing a summary of the stages and subtypes per cohort. Personally, I would not randomly assign the samples because now it is possible that stages and subtypes are not included in a certain cohort. Too few LCNEC and SCLC patients were included in the study for subtyping.

Line 122 ‘…, we also included an external independent test cohort of 142 participants, …’: I am missing a summary of the clinical information for these patients.

Line 144 ‘PCA demonstrated clear group separation.’: Rather a visible clustering between lung cancer and healthy individuals.

Line 186 ‘We used 110 biologically significant features as input to construct the machine learning model, including 100 TSS sites that were higher in cancer samples than in healthy individuals and 100 TSS sites that were higher in healthy individuals than in cancer samples.’: The numbers do not match.

Line 203 ‘… to verify the model's performance.’: For the sake of clarity, I would add ‘in distinguishing between cancer and non-cancer samples.’

Line 255 ‘model’s efficacy in predicting EGFR mutation status…’ (also line 285): Knowing the mutation status alone is not enough; it is still important to know the specific mutation for example the presence of an EGFR T790M variant is generally associated with a poor response to 1st-2nd line EGFR tyrosine kinase inhibitors in lung adenocarcinoma, but a good response on 3rd line EGFR tyrosine kinase inhibitors including Osimertinib.

Line 339-340: No second centrifugation to ensure that no white blood cells are transferred.

Line 342 ‘An appropriate amount of blood sample’: What is an appropriate amount? 4 ml of Plasma?

Line 353 ‘… dry for 10-15 minutes.’: This is rather long, Overdrying the beads may lead to reduction in yield.

Line 357 ‘After the construction of each sample library is completed, …’: Please add library construction details (input amount in ng, kit used, any adjustments to the protocol).

Line 378 formula: Where does the '150' come from?

Comments on the Quality of English Language

Line 341 ‘…, sample extraction is performed …’: For the sake of clarity, I would replace ‘sample extraction’ by ‘cfDNA extraction’

Line 342 ‘An appropriate amount of blood sample’: For the sake of clarity, I would replace ‘blood sample’ by ‘plasma sample’.

Author Response

Dear reviewer,

Thank you for your time.

Reviewer 2 Report

Comments and Suggestions for Authors

The article is potentially interesting, but it appears to be a pilot study that needs to be confirmed in a larger sample population.

The results showing ROC curves with such high significance need to be confirmed. As there is no gold standard for comparison, a large number of patients/healthy volunteers need to be analysed. As this is a proof-of-concept study, this must be made clear in the title and also emphasised in the discussion.

 A major problem concerns the cohort of the study, which is not well described. For example, there is no information on the age and gender of the participating patients. The same applies to the cohort of healthy volunteers. Another problem is the fact that the authors have not shown whether age influences the results obtained. This point needs to be clarified.

The recruitment of the healthy patients (and how they were recruited) and their age and gender data need to be described in detail. Was informed consent obtained for them? This must be added. In addition, the authors must compare the patients with the controls and show that age and sex are similar in both groups.

As many patients had high-grade lung cancer, the authors should analyse the correlation of overall survival with DNA analysis.

The difference in gene expression between stages I/II and stages III/IV should be analysed and described.

Selected tissue samples need to be analysed for gene expression to compare the results with those of liquid biopsy.

The limitations of the study and future directions need to be outlined in the discussion section.

The methods in section 4.2 are not well described and information about the providers of the instruments and reagents is missing. In addition, the description of DNA extraction needs to be carefully revised. For example, it is unnecessary to describe how many times the solution in the tubes was inverted to mix the reagents and then not describe the volumes with words such as “appropriate” or explain what some acronyms mean (e.g. GDA, PWD, etc.). Another example is the storage conditions that must be specified.

In the case of blood sampling, it is unclear when it was performed: at the first visit, when the diagnosis was communicated? This is an essential point that needs to be specified in the methods.

Most of the methodology for data analysis is described in the Results section and could be moved to the Methods section.

The quality of Figure 2D is very poor and needs to be improved.

There are many typos such as “mutaion”

The legend in the tables and figures needs to be revised. A description of all abbreviations used here is missing.

Acronyms such as "TSS" should be avoided in the title.

Author Response

Dear reviewer,

Thank you for your time.

Round 2

Reviewer 1 Report

Comments and Suggestions for Authors

Dear Authors,

Thank you for thoroughly addressing my comments.

I think the new title fits the manuscript better and it is now clear that one of the objectives was to try to predict the EGFR mutation status.

Concerning the latter, I do not agree with line 311-312 in your discussion ‘Compared to traditional methods of molecular typing based on tissue samples, our non-invasive plasma cfDNA TSS mutations prediction approach achieves similar clinical application effects.’. You have added information about the types of EGFR mutation in Supplementary Table S1 but additional molecular typing, identifying the specific mutation, is still necessary for treatment decision.

This is also what I meant by 'lacks tumour-specific information for treatment response' and not information about treatment response. I am sorry this was not clear to you.

I find it valuable that you added the advantages of the method compared to others in a clinical setting, but I still wonder how the sequencing cost compares?

Author Response

Dear Reviewer,

Thanks in advance.

Reviewer 2 Report

Comments and Suggestions for Authors

Authors must complete their article by adding figures R1, R2 and R3 as additional figures and clearly indicating this in the text. Furthermore, they need to emphasise that they have not investigated the concordance of results between plasma and tissue samples, which is a limitation of the present study. Finally, Table 1 must be considered as clinical information from lung cancer patients and not from lung cancer samples. Supplementary tables with clinical characteristics of patients with age and sex and healthy subjects (age and sex) must be included in the results section and cited accordingly (e.g. information on patients and healthy cohorts are in...)

Author Response

Dear Reviewer,

Thanks in advance.
